# Does University–Industry Engagement Assist Women in Generating Business Income in Emerging Economies? Evidence from Malaysia

**Eni Murdiati [1], Jawazi Jawazi [1], Nor Balkish Zakaria [2,*] and Kazi Musa [2]**

[1] Faculty of Da'wah and Communication, Universitas Islam Negeri Raden Fatah Palembang, South Sumatera 30126, Indonesia; enimurdiati_uin@radenfatah.ac.id (E.M.); a.jawazi@radenfatah.ac.id (J.J.)

[2] Accounting Research Institute (ARI), Universiti Teknologi MARA, Shah Alam 40450, Malaysia; kazimusa1@gmail.com

[*] Correspondence: norbalkish@uitm.edu.my

**Abstract:** Women have a significant role in economic development in emerging economies like Malaysia through employment and business ownership. This is crucial for the family's survival and the prosperity of local economies—especially during an economic crisis or turmoil. Even though SMEs contribute to the local economy, their long-term viability is always uncertain. On the other hand, universities are considered indispensable and requisite contributors to the survival of this SME. This study, hence, evaluates the performance of women-owned small and medium businesses in conjunction with university–industry collaboration. Based on stratified sampling, a questionnaire survey was undertaken among regional SME female owners from various industries closely located to university campuses across different states post-pandemic in June 2021. The 227 female-owned SME responses showed a strong correlation between the university's contribution to SMEs' revenue. The findings demonstrate the importance of university support in marketing and training to SMEs' revenue according to women business owners. These findings accentuate the need for institutional policymakers to generate more profound knowledge and the need to keep ancillary longitudinal initiatives for sustainable business performance, especially among women, via university–industry arrangements.

**Keywords:** women business; income sustainability; small medium enterprise (SME); university–industry engagement



## 1. Introduction

Women are considered effective business owners because they have a strong drive, desirable characteristics, and aptitudes for vigorous economic growth (Khan et al. 2021). Due to the significant role that women have played in economic progress, the auxiliary factors to their success are valuable for observation. Women-owned small firms were the fastest-growing sector during the 1980s and 1990s (Loscocco and Bird 2012; Hussain et al. 2021). Throughout the 1990s and 2000s, the proportion of women-owned firms grew more quickly than that of men-owned businesses. Women-owned enterprises expanded twice as quickly as male-owned businesses between 1997 and 2007. The number of women-owned businesses among all U.S. non-farm firms increased to 28.2 per cent by 2002, reaching over 30 per cent by 2007. Compared to men's, women's business ventures exhibit several distinct traits (Rafiki and Nasution 2019). As a result, women's entrepreneurship attracts a unique area of study.

Even among successful small business owners, women make less money and sell less than their male colleagues (Sayre et al. 1991). The reasons why women are generally at a disadvantage are carefully looked at. Differences between the owners and the small businesses can explain the differences in business performance. The size of women's businesses is the most critical factor in explaining this difference. The gender gap is also

affected by the fact that women do not have as much experience and tend to work in industries that do not pay. Even successful women are worse at taking advantage of business opportunities than their male counterparts (Loscocco et al. 1991) because of the structural disadvantages they face inside and outside the business world. How a female small business owner makes sales and money is the same as how a male small business owner goes about it.

Although many women entrepreneurs are successful, their true potential has yet to be realised. Numerous explanations have been proposed, ranging from societal beliefs about women's traditional responsibilities as homemakers to the lack of equal opportunities made available to women, resulting in a market failure that inhibits women from reaching their full entrepreneurial potential. Furthermore, it has been asserted that women entrepreneurs lack confidence and stamina (Kirkwood 2009), placing greater emphasis on family and work–life balance.

According to Mansor et al. (2023), the population of Malaysia in 2022 was 29.62 million; out of that, 3.2 million Malaysians were employed. Males constituted 61% of business owners, whereas the female gender only contributed 39%. Approximately 400,000 Malaysian women are still unemployed.

This phenomenon may inspire women to pursue significant entrepreneurial endeavours for developing themselves and their families. In addition to the current adversity, the younger generation chooses entrepreneurship because it can provide a lucrative salary. Mansor et al. (2023) report that the value created by women-owned businesses is 4.5% (MYR 46.53 billion), whereas the gross production value is 3.7% (MYR 92.4 billion).

### 1.1. Women in Business

Small-scale businesses, especially among women, are vulnerable to any turmoil. During the pandemic, small and medium enterprises have been among the most brutally hit, initiating alarm in the industry, especially among young and less resilient entrepreneurs. Hence, female entrepreneurs were also at risk. The shockwaves of the crisis have affected about 92% of young entrepreneurs (Malaysian Reserve 2021), from lockdowns and declining demand to damaged supply chains and finance constraints. More Malaysian women turned to entrepreneurship during the pandemic because their families' finances were in a slump.

The Malaysian Ministry of Entrepreneur Development and Cooperatives and the Ministry of Women, Family, and Community Development have set aside MYR 18.2 million for programmes and assistance released by Department of Statistics Malaysia in 2021 exclusively for women and disabled company owners to aid the target groups, particularly in the post-COVID-19 recovery, as well as help them to expand their businesses.

From another viewpoint, small- and medium-sized businesses (SMEs) have rapidly expanded in Malaysia over the past few years, contributing 39.1% of the country's GDP in 2021. In Malaysia, 9031,332 SMEs totalled 98.7 percent of all company enterprises, spanning all sizes and industries reported by the Economic Census Malaysia in 2012, Profile of Small and Medium Enterprise). In Malaysia, SMEs primarily operate in the service, manufacturing, construction, and agricultural industries as well as in mining and quarrying reported by the Department of Statistics Malaysia. Small and medium-sized businesses (SMEs) must survive to maintain a robust business environment and economy (Zakaria 2009).

Mansor et al. (2023) state that different states have different business concentrations of SMEs. Selangor (19.8%), Federal Territory of Kuala Lumpur (14.7%), Johor (10.8%), Penang (7.4%), and Sarawak are the states with the highest percentages of SMEs (6.7 percent). Additionally, numerous government initiatives and programmes have supported the founding and growth of SMEs since the 1970s (Saleh and Ndubisi 2006). Like Thailand, SMEs play a significant part in developing the country's economy, and this, in turn, helps the healthy development of the economy in Malaysia (Chittithaworn et al. 2011). Additionally, SMEs

are anticipated to play a significant role in economic development, job generation, and developing nations' transition (Gunto and Alias 2013).

*1.2. Small Medium Enterprises*

Many factors affect SMEs' growth and ensure the company's sustainability. According to Lönnqvist (2002) and Williams (1998), both tangible (measurable) and intangible (not immediately measurable) variables contribute to an enterprise's success. Soemitra et al. (2022) elaborated on these factors. Networking, products, the capacity to concentrate on markets and customers, the development of new products, financial performance, objective and subjective market approval, attractiveness, and product quality are a few examples.

Additionally, not all SMEs in Malaysia must have their annual returns audited. Hence, there will be no financial check and balance on the performance. Their ability to continue operating is also uncertain (Azmi et al. 2013), particularly by the excellence auditors.

Small- and medium-sized firms have no worldwide definition (Khalique et al. 2011). The total figures of full-time personnel and annual sales volume were also utilised to identify SMEs in Malaysia. The SME Corporation Malaysia classified medium enterprises as those having 50 to 150 full-time employees in 2008. On the other hand, small businesses are described as having between 5 and 49 employees. Micro-enterprises are companies with at least five employees. These SMEs are small enterprises, microbusinesses, and medium-sized businesses.

Since the definition and the criteria of SMEs vary between countries, it is required to discuss the definition and specifications of SMEs from the context of Malaysia. A revised SME definition was approved at the 14th National SME Development Council (NSDC) Meeting in July 2013. The concept includes all industries, including manufacturing, agriculture, construction, mining and quarrying, and services (SME Corporation Malaysia 2008). The two criteria used to determine the definition with the "OR" basis are the number of full-time employees and sales turnover as follows: SMEs are organisations in the manufacturing sector that have a full-time staff count of no more than 200 or a sales turnover of no more than MYR 50 million (Cheah et al. 2022; Vrchota et al. 2019). SMEs are companies with annual sales turnover or the number of full-time employees under MYR 20 million in the services and other sectors.

In Malaysia, 98.5 percent of businesses and enterprises are regarded as SMEs after fulfilling these requirements (Alam et al. 2022). This industry's GDP contribution was MYR 491.6 billion in 2017 and MYR 521.7 billion in 2018 (Alam et al. 2022). Additionally, the SMEs supported exports worth MYR 166.2 billion in 2017 and MYR 171.9 billion in 2018. Furthermore, a significant number of jobs were also created in this industry; the number of active workers in this sector as a share of the total labour force was 65.3% in 2016 and 66.00% in 2017 (Omar et al. 2022).

Based on research in a South Korean setting about the effects of entrepreneurship on SMEs, Yi et al. (2018) concluded that process innovation positively impacts marketing competence concerning the development of new products. From the perspective of SMEs in Indonesia, Sumiati (2020) also underlined the crucial connection between entrepreneurial and innovative aspects. The study's findings emphasise the significance of innovation for SMEs in raising their level of quality and keeping them competitive with other businesses. It has been discovered that an enhancement in business reputation is positively correlated with industry innovation, and public outreach, in particular colleges, would raise corporate sustainability in the long run.

*1.3. University–Industry Collaborations*

One major component that contributes to the growth and success of businesses is the community, which constitutes a sizable portion of their clients. As a community constituent, the university is connected to this community. Generally, universities and higher education institutions operate as a fully connected community relying on localised, national, and global societies (Myzrova et al. 2023; Cortese 2003). In addition to the

"knowledge economy," the university connects industry and community members, making it one of the most significant contributors to local prosperity growth (Acworth 2008). Arena et al. (2018) highlighted that universities are considered research centres seeking to collaborate with industries to create innovative products. This explains the value of a university's appearance in the surrounding community, especially in helping businesses.

However, Hatipoğlu (2021) argued that universities must fully utilise their vital position to contribute to the ecosystem's improvements. More can be achieved by integrating the programs with teaching and research and increasing their specialisation in various social issues. Azároff (1982) posits that industry and university cannot get along; the industry feels that the university is unreliable, while the university thinks the industry is too restrictive.

The benefits of connections and cooperation between businesses and universities, including training and talent hiring, are highlighted by Nguyen and Nguyen (2020). The process variables, contextual, organisational, cooperation, and benefit, were especially examined from the perspective of tourist development. It was discovered that the first two were the most crucial for successful cooperation between the two sides. When businesses are led by university expertise, crucial company performance and default risk are also evident (Zakaria 2012).

The distance between companies, communities, and universities is a long-standing topic of discussion. The university and its surrounding communities need to be connected. The relationship between academic institutions and business is nothing new (Etzkowitz 2001; Feldman and Desrochers 2004), but the specific ways in which it translates into quantifiable results, particularly for the community, are still thought to be novel (Philpott et al. 2011; Anatan and Nur 2023). The contributions made by universities to societies, particularly in helping them to produce revenue and extend their lifespan, need to be sufficiently supported by empirical data from earlier studies.

To support the need for university and industry collaboration, Louis et al. (1989) and Klofsten and Jones-Evans (2000) highlight the range of entrepreneurial activities divided into hard and soft initiatives. In comparison to softer industries (academic publishing, grant writing, and contract research), which are sometimes perceived to have minimal entrepreneurial value, challenging activities (such as patenting, licencing, and spin-off firm formation) are typically viewed as having more measurable outcomes (Anatan and Nur 2023; Rasmussen et al. 2006).

However, university–industry collaboration faces several challenges. How effectively is a university contributing to society, especially small businesses and female business owners? If so, in which area does the contribution underlie?

The fact that most people, especially small business owners, need to learn more about intellectual property makes it hard for the two to work together. Academic institutions help the economy and the community by carrying things out like patenting, which comes from scholarly ideas. However, universities and small business owners must discuss to understand these business fundamentals better while generating small royalties (Anatan and Nur 2023; Philpott et al. 2011). Even though academics have more practical ways to share their knowledge with industry and the community, such as through publications, conferences, consultations (Cohen et al. 2002), initiatives to improve academics' entrepreneurial knowledge and experience (Myzrova et al. 2023; Rahim et al. 2015), attachments, certified partners accreditation, and a dedicated innovation programme, the gap has lowered the community's academic expectations of academics. Scholars at universities tend to care more about their networks and colleagues than about the communities around them.

A few sections have been constructed for this study that examine how university contributions affect SMEs' revenue, namely in marketing, consulting, human resources, and training among women entrepreneurs in conjunction with the university–industry collaboration. The review of previous research that discusses the results in the research field comes after this section. The methodology section and the sampling strategy highlight the SME groupings, respondents' localities, and relevant assessments. Findings and discussions

regarding the gathered information are described in the following section. The overall study is then concluded in the final section.

## 2. Literature Review and Hypothesis

Women's entrepreneurship is a global issue that has gained much scholarly attention over the years (Henry et al. 2016; Nguyen 2022). It is an excellent focus for concerted scholarly work since it improves economies regarding employment creation (Bosma and Levie 2010) and is also recognised as a foundation of trade diversity in several socioeconomic circumstances (Verheul et al. 2006). Although women have contributed significantly to entrepreneurship, several impediments impede their achievement as entrepreneurs (Torres-Ortega et al. 2015).

In contrast, in many developing countries, women entrepreneurs still need to receive the necessary support to launch businesses (Roomi and Parrott 2008). Sadly, notwithstanding their consistent contribution to GDP (Bosma and Levie 2010) and poverty alleviation, less consideration has been paid to women capitalists in emerging market thrifts (Khan 2014) as a result of the complex confluence of religious, familial, and social variables (Roomi 2013). Due to sexual identity power systems built on inequity and prejudice, women endure gender-based discrimination (Roomi et al. 2018).

Accomplishment is a secret motivator developed through human sensory perception (McClelland et al. 1976). While Jayeoba et al. (2013) define it as a source of inspiration for lengthy entrepreneur success and an indicator of the improvement of the desire to obtain a significant achievement in his life or business, it is characterised as the desire for success or excellence (Balogun et al. 2017). Like accountability and focus on achievement, the desire to succeed necessitates difficult labour (Rauch and Frese 2007).

Nurwahida (2007) identifies achievement as one of the necessary characteristics of most of the highly successful female business entrepreneurs as well as motivated and risk-takers. Several studies have investigated how highly motivated, entrepreneurially minded, and management-skilled top female businesspeople can boost their business success. Additionally, this motive supports senior managers' efforts to exhibit objective behaviour (Rasheed 2001). As per Dolan et al. (2008), inspirational behaviour not only aids managers but also acts as the basis for their goal achievement.

However, several Malaysian SMEs need help understanding branding, marketing tactics, and customer loyalty, resulting in a competitive disadvantage and the loss of exceptional chances (Alam et al. 2011). Recent research conducted by Ambad et al. (2020) outlined several obstacles that impede the success of SMEs, including budgetary restrictions, supplier issues, human resource problems, an absence of creativity and innovation, and brand marketing concerns. Therefore, it is widely believed that fostering university–industry cooperation is the best strategy to support SMEs in overcoming those obstacles.

Asri et al. (2020) posit the significance of exceptional packaging among SMEs' products, particularly food and drink, in luring prospects and stimulating the desire to make repeat purchases. Women-owned small businesses need additional support since they are generally underrepresented in Asia, particularly Malaysia. Additionally, there are not many enterprises, including SMEs, held by women in Malaysia, but they are growing. Therefore, universities must support SMEs run by women to promote gender equality, inclusive development, and, most significantly, the development of women.

In a nutshell, this is one of the potential assistances that university engagement can provide via marketing programmes to SMEs. Second, universities highly prioritise women's development, and supporting women-owned businesses is a crucial channel to support this objective. Third, promoting economic growth is among the university's objectives when working with SMEs, particularly SMEs owned by women; therefore, informal learning has been seen as an essential part of Malaysia's government's agenda on economic growth and women's development. Each public university must now have an entrepreneurship centre, which institutions' administrators frequently assess by evaluating their targets (Ngah 2018). University marketing reflects a university's programme that helps promote overall SMEs

and SMEs owned by women to fulfil their staff and students' needs, for instance, weeks to meeting potential customers, food trucks carnivals, and meet and greets on selected days in a month open to smaller business owners. Hence,

**H1.** *There is a significant association between women's small business revenue and university marketing.*

Temel et al. (2013) indicated that university engagement and innovation have a substantial impact on raising the profit of SMEs when the level of dedication and effort put forth is at its peak in the context of Turkey. This demonstrates that most firms find accepting indecent proposals or academic commitments unacceptable, especially in the tight market competition. In other terms, the institution must plan and implement the contribution it wants to make even while seeming relevant to the business world. Despite their financial successes, SMEs in the neighbourhood only see universities as social benefactors.

Using a multilayer approach, Messeni Petruzzelli and Murgia (2021) demonstrated the critical impact of SMEs' absorptive capacity and social and geographic proximity amongst their collaborating firms. Additionally, their findings demonstrated the benefit of regional knowledge spillovers near the technological domains where the inventions emerged. Their research helped researchers better understand interactive learning in university-SME R&D partnerships and how knowledge spillovers from SMEs' Regional Innovation Systems may support it (RIS).

Abdullahu and Masrom (2017) provide some key explanations for the increased university–industry engagement and university–women-owned enterprise linkage by outlining the perceived advantages of Malaysian university–industry collaboration. Othman (2011) and Abdullahu and Masrom (2017) assert that because universities help the industry by providing the human capital it needs to operate, they are unquestionably helpful to growing women-owned businesses and SMEs. Additionally, Othman and Omar (2012) discovered a significant gap in university–industry involvement, particularly at the micro level, where both parties needed to be more competent to cooperate. This may result from their lack of exposure to the potential consultation and contributions they may be able to make. University consultation reflects the free university programmes that are carried out to help small-scale businesses learn the theoretical business from academic perspectives, for instance, simple book-keeping consultants, basic costing and pricing seminars, and business procurement innovations for smaller business owners. Therefore, the women owners and CEOs of SMEs can take advantage of these benefits at a low cost or for free from the universities to grow their business. Consequently, this study hypothesises that

**H2.** *There is a significant association between women's small business revenue and university consultation.*

According to McClelland's (1988) motivation theory, there are three distinct motivational criteria for human performance: the need for accomplishment, the demand for dominance, and the desire for allegiance. To achieve their objectives, senior managers need to desire success (Dewi et al. 2016; Rozzani et al. 2016). SMEs remain well recognised for their contribution to economic development since they manage several sectors of the manufacturing sector, including the processing and production of raw materials such as food, beverages, rubber, and wood, among others. In 1991, according to data compiled by the Malaysian Ministry of International Trade and Industry (2019) (MITI), SMEs contributed 20% of the manufacturing GDP. Kassim and Sulaiman (2011) presented additional data to substantiate the figure, saying that SMEs now account for 3.12 percent of total manufacturing sector employment, where many owners and workers are women and are predicted to grow substantially over the next few years.

Even though it is well known that SMEs create many jobs and contribute a lot to sustainable development and women's development, the biggest challenge for education providers is to match graduates' skills and knowledge with open jobs. If the higher education system in management cannot meet society's future needs, the nation's social,

economic, and environmental conditions will worsen. So, for long-term sustainability, management graduates who will be managers in the future should obtain a good education that gives them the skills and knowledge they need. This can be carried out through a sustainable curriculum that connects SMEs while still in school. This can be a stepping stone for business start-ups, job opportunities, and industry intakes (Shivany 2021). At the same time, this process can motivate girls and women in schools and universities to engage in business or be the CEO of their own formed businesses.

Engagement between universities and industry is closely linked to scientists' employment by businesses, consultancies, student industrial training programmes, marketing courses, information transfer efforts, and other pertinent initiatives (Yee et al. 2015; Rozzani et al. 2016). The researchers also stated that mutual trust in one another's abilities to benefit one another is essential for a successful university–industry interaction. To create good trust and commitment, Lavie et al. (2012) advised that the institutional and business companies engaged in university–industry partnerships respect and value one another's routines and cultures.

Nevertheless, Malaysia is one of many nations that supports university–industry collaboration through SMEs. According to Mitanoski et al. (2013), SMEs are the primary producers of new jobs and a force in the fight against unemployment. They have also emphasised that a solid partnership between academia and business serves as a resource for current recruiters and prospective job seekers. University human resources reflect the opportunity for employment that the university may offer to the surrounding businesses and nearby businesses owned by women, for instance, part-time tutors, part-time markers, research assistants, and casual workers, to free up the time of female business owners that may need to add more funding that could increase their small-scale business revenue. Thus, this study hypothesises that

**H3.** *There is a significant association between women's small business revenue and university human resources.*

To be more relevant and forward-thinking in servicing the industry, Hamdan et al. (2011) claim that universities must move beyond their conventional tasks of only generating graduates. The interaction between industry and universities can substantially aid Malaysia's economic development. According to studies, Malaysia's thriving economy can be attributed to close university–industry collaboration. These programmes can result in a wide range of advantages for knowledge transfer, research and development, and training for commercialisation.

Cooperation between the university and the industry is required to improve the committee's skills and revenue. Salleh and Omar (2013) devised a model of three essential elements, universities, governments, and industries, for universities and industries to engage successfully. At the same time, female-owned businesses have the potential to thrive faster in Malaysia under the support of such benefits provided by the universities.

University training reflects the free university training programmes that are carried out to help small-scale businesses via teaching–learning pedagogies, including branding creation, label for marketing training, female business personalities and grooming, and interpersonal and intrapersonal business coaching. With this coaching and mentoring offered during training, female business owners could help to increase their small-scale business revenue. Therefore,

**H4.** *There is a significant association between women's small business revenue across university training.*

### 3. Methodology

As one of Malaysia's public higher education institutions, Universiti Teknologi MARA (university) is a large institution with numerous faculty members. There are 35 campuses, with at least 1 in each state (except the Federal Territories). As of December 2020,

the university is home to approximately 170,000 students and more than 17,000 faculty and workforce.

University Transformation Division (2020) Malaysia has made this university the largest institution of higher education in Malaysia. Most UiTM campuses are located on the outskirts of cities or in rural regions. The faculty and students reside either on campus or in nearby residential neighbourhoods. In either instance, the university society is a group that adds to the dynamics of the greater community in which they live. In some ways, the placement of university campuses helps the expansion and achievement of local small and medium companies (SMEs) because the institution's neighbourhood is also a potential customer for these firms. As a result, choosing UiTM as the research site for this study is crucial, given that its physical appearance encompasses all Malaysian states besides the size of the population across students and academic and non-academic staff.

Local and close small- and medium-sized enterprises (SMEs) surrounding the university campuses were surveyed within a 20 km radius of UiTM branches in Johor, Melaka, and Negeri Sembilan states, targeting the women business owners. The 20 km radius is a practical and strategic choice for surveying local SMEs, particularly women-owned businesses, around university campuses. It balances the need for proximity, accessibility, and a manageable research scope while ensuring that the study is relevant to the university's local community. Almost 600 questionnaires were delivered to each branch using stratified sampling procedures in June 2021 to reflect the post-pandemic effect. The survey forms were then sent by hand to the identified female-owned small businesses within a 20 km radius of each university branch campus. Their identifications were made through their business listing in the local municipal business registrations. Additionally, the study confirmed the SMEs' connections to the selected institutions from the database of local governments as well as from the university's industrial support services during the early stage of sample selection. The SME category includes single proprietorships, partnerships, and corporations. A total of 227 survey forms were finally returned, collected, and analysed for future examination. This made up a 37.85% response rate (227 out of 600).

A descriptive analysis and bivariate and multivariate tests were undertaken to serve the study's ultimate objectives. A hypothesis was established to examine the relationship between SMEs' revenue and university contributions to SMEs, including marketing, consulting, human resources, and training, while controlling for ownership type, employee number, and the years of business establishment. For this, a model was developed.

$$REV = \beta_0 + \beta_1 MCON + \beta_2 CCON + \beta_3 HRCON + \beta_4 TCON \\ + \beta_5 BOWN + \beta_6 STAFN + \beta_7 AGE + \epsilon \tag{1}$$

where

REV = business gross income generated yearly in MYR.
MCON = marketing contribution made by university to business (1 = low, 2 = medium, 3 = high, 4 = very high).
CCON= consultancy contribution made by university to business (1 = low, 2 = medium, 3 = high, 4 = very high).
HRCON = human resource contribution made by the university to business (1 = low, 2 = medium, 3 = high, 4 = very high).
TCON = training contribution made by university to business (1 = low, 2 = medium, 3 = high, 4 = very high).
BOWN = type of business ownership (sole proprietor, partnership, or sdn bhd at private limited.
STAFN = staff number currently hired.
AGE = number of years the company has been in operation.

## 4. Results and Discussion

Based on Table 1, Pearson correlation coefficients demonstrate the link between pairs of variables at 95% and 90% confidence levels. The coefficients indicate a positive and substantial relationship between SME revenue and business age. In terms of university contribution to these SMEs, marketing and pieces of training have a favourable and significant association with the income of these SMEs. The number of SMEs hired is also positively related to the marketing supplied by the university. The business age is beneficial and significant to the university's influence, marketing, and training.

**Table 1.** Pearson Correlation Results.

|  | REV | STAFN | AGE | BOWN | MCON | CCON | HRCON |
|---|---|---|---|---|---|---|---|
| STAFN | 0.123 * | | | | | | |
| AGE | 0.142 ** | 0.113 | | | | | |
| BOWN | 0.113 | 0.104 | −0.103 | | | | |
| MCON | 0.124 * | 0.119 * | 0.129 * | 0.105 | | | |
| CCON | 0.114 | 0.107 | −0.105 | 0.102 | 0.106 | | |
| HRCON | 0.106 | 0.114 | 0.110 | 0.113 | 0.114 | 0.105 | |
| TCON | 0.127 * | 0.113 | 0.125 * | 0.124 * | 0.115 | 0.116 | 0.12 |

Note: ** significant at 95% and * significant at 90%.

Revenue is the SMEs' yearly gross income in MYR. University contribution perception through marketing, consulting, human resources, and training to these SMEs was measured on a sliding scale; 1 is low, 2 is medium, 3 is high, and 4 is very high. The three types of business ownership are sole proprietorship, partnership, and private limited company. STAFN is the number of employees hired, whereas Age is the number of years the company has been operating.

To examine how university contributions affect SMEs' revenue in marketing, consulting, human resources, and training among women entrepreneurs, a regression analysis was performed in addition to the bivariate Pearson correlation results while controlling for the effect of SMEs' ownership type, staff number, and business age.

Table 2 depicts the results of our focused variables from the multiple regression. The coefficient for MCON is 0.169, and the T-statistic is 2.34 (t = 2.34 > 1.645), which is statistically positive and significant at the 90% confidence level. It implies that a 1-unit increase in MCON (marketing contribution by the university to business) is associated with a 0.169 percent increase in the revenue of small and medium enterprises run by women while holding other variables constant. Vodă and Florea (2019) state that university marketing support helps to grow businesses owned by women in the Romanian university context. Kamberidou (2020) states that female company owners still have to deal with the multitasking vortex, a lack of funding, marketing expertise, support services, and limited access to business networks, technology, and digital markets. As a result, the author admits that marketing, training, and loan facilities encourage women-owned businesses.

At the same time, the coefficient for CCON (consultancy contribution by the university to business) is 0.030 (t = 0.412 < 1.645), and it does not appear to be statistically significant. This means consultancy contributions made by the university to businesses owned by women barely impact the generated income. Though consultancy is an important factor for any business, it provides neutral results. Minimal consultancy support from the university can cause it. Martinez Dy and Jayawarna (2020) and Hutchings et al. (2020) claim that businesses with external and internal consultancy support help to find new buyers, raw materials, other logistics, financial sources, and other collaborations for women-owned enterprises. This means that Malaysian women-owned SMEs need more consultancy support to spur rapidly.

**Table 2.** Multiple Regression Results Among the Independent Variables.

| Variables | Coefficients | T Stat | VIF |
|-----------|--------------|--------|-----|
| REV | | | 1.107 |
| MCON | 0.169 | 2.34 * | 1.120 |
| CCON | 0.030 | 0.412 | 1.169 |
| HRCON | 0.044 | 0.697 | 1.280 |
| TCON | 0.159 | 2.13 * | 1.090 |
| BOWN | 0.025 | 0.336 | 1.184 |
| STAFN | −0.058 | −0.802 | 1.167 |
| AGE | 0.210 | 2.85 ** | 1.242 |

Note: ** significant at 95%; * significant at 90%.

Table 2 also demonstrates that the coefficient for HRCON (human resource contribution by the university to business) is 0.044 (t = 0.697 < 1.645), which is statistically insignificant. Neumeyer et al. (2019) found in Florida, USA, that SMEs run by women entrepreneurs can make much more revenue if universities support them with human resources. So, UiTM Malaysia might have limited efforts in HR support for SMEs run by women in UiTM's surrounding areas. Therefore, Malaysian universities should deploy more human resources for the SME sectors of Malaysia, at least in the university's surrounding areas, to support overall and women-owned SMEs.

The coefficient of TCON (training contribution by the university to business) is 0.159, and the T-statistic of 2.13 indicates that TCON is statistically positive and significant at the 90% confidence level. The results indicate that a 1-unit increase in TCON support from the university is associated with a 0.159-unit increase in profit generation while holding other variables constant. Setini et al. (2020) also found identical results in women entrepreneurs in Bali, Indonesia. As the training contribution by the university encourages women-owned SMEs, universities in Malaysia should continue this support for further sector growth.

The coefficient for BOWN (type of business ownership) is 0.025, and it does not appear statistically significant. When universities encourage women's businesses, business types hardly play a significant role in helping women's businesses thrive in the context of SMEs near UiTM. The coefficient for STAFN (staff numbers currently hired) is −0.058, and it does not appear statistically significant. The STAFN results contradict several pieces of research, although they may not matter because most of the SMEs were shut down by the COVID-19 pandemic in Malaysia and most of the staff stayed at home (Cooke and Xiao 2021).

The coefficient for AGE (number of years the company has been in operation) is 0.210, with a T-statistic of 2.85; this indicates that AGE is statistically significant at the 95% confidence level. It indicates that increasing an SME's age generates more revenue. Usually, when a business works for many years, its experience, skills, product quality, and marketing become more stable and stronger, supporting profitability (Zeitun and Goaied 2021; Mallinguh et al. 2020). Based on the provided results, MCON, TCON, and AGE are statistically significant with regard to income generation in SMEs run by women in the radius of UiTM in Malaysia. At the same time, CCON shows insignificant results, meaning that universities have minimal efforts or some limitations in providing enough support in consultancy to SMEs.

So, at a 90% confidence level, the results suggest that SMEs' revenue is significantly affected by the marketing and training supplied by the university. According to Zakaria (2009), earnings and cash flow can provide corporate liquidity performance in Malaysian small and medium firms. As a result of good income performance, marketing techniques may be improved, and more training could be supplied. Furthermore, at a 95% confidence level, business age indicates the importance of association to these SMEs' income.

Thus, the findings supported hypotheses H1 and H4 that there is a significant relationship between SMEs' income and university contribution in terms of marketing and training delivered. The results indicate that the marketing services given by the university and the training provided to the surrounding SMEs have assisted them in increasing

their revenue. Business age impacts revenue since the more established the business, the higher the revenue. The results of the variance inflation factor (VIF) reveal that there is no multicollinearity problem in this study parameter.

This study found that university marketing and training programmes help increase smaller-scale business revenue among females. Arena et al. (2018) support the finding of this study in the context of the technology transfer process (could also be regarded as part of training in this current study context). Universities can also assist women-owned SMEs by recommending new funding sources and marketing opportunities, creating business networks, implementing cutting-edge technology, and undergoing digital transformations (Jamil 2021; Maseda et al. 2022). However, this is contradicted by Azároff (1982), who posits that industry and university must get along better. While the university is unreliable, the industry needs to be more relaxed.

## 5. Conclusions

This study aims to investigate the effects of university contributions on the marketing, consulting, human resource, and training activities of SMEs among women entrepreneurs. The parameters of the study also regulate the impact of the type of ownership, the number of employees, and the age of the SME.

Establishing university campuses around the country is crucial in fostering the growth and success of local SMEs since the institution's community also patronises these companies. However, imparting knowledge and experience to the neighbourhood remains a problem due to the grave concern about the institution's distance from the community. Considering the above, this study was carried out to gather information on the revenue factors of women SME owners under the influence of university contributions—marketing, consultancy, HR, and pieces of training—while controlling for respondents' ownership type, employee number, and business age.

This study presents twofold theoretical and practical implications. Theoretically, university contributions to small businesses owned by women have critical theoretical implications because they can help remove some of the most significant barriers to economic growth and help make society fairer and more sustainable. Discrimination and inequality based on gender are significant obstacles to economic growth, and addressing them is crucial to achieving sustainable development goals. Contributions from universities can help close the gender gap by giving women who own small businesses access to resources, training, and networks that are usually close to them. University contributions to female small businesses can have important practical implications, as they can assist these businesses to overcome some of the critical problems they encounter and become more profitable, sustainable, and influential. Access to finance and financial resources that may be challenging to obtain can be made possible through university programmes and efforts that promote female small business owners. In addition, a university can give female small business owners access to mentoring, training, and educational programmes that help improve their knowledge and abilities. Female small company owners can gain more visibility with access to networks and resources.

This study was only limited to Malaysia-selected universities and small (SMEs) female business settings. Future research could examine other geographical locations with some traditions that may limit female business endeavours based on different or specific cultural requirements.

**Author Contributions:** Conceptualization, E.M. and N.B.Z.; methodology, N.B.Z.; software, J.J.; validation, E.M., N.B.Z. and K.M.; formal analysis, J.J.; investigation, N.B.Z.; resources, E.M.; data curation, E.M. and J.J.; writing—original draft preparation, E.M. and J.J.; writing—review and editing, K.M.; visualization, N.B.Z.; supervision, N.B.Z.; project administration, N.B.Z.; funding acquisition, E.M. and J.J. All authors have read and agreed to the published version of the manuscript.

**Funding:** The authors would like to thank the support from the Universitas Islam Negeri Raden Fatah, Palembang, Indonesia and Accounting Research Institute HICoE of Universiti Teknologi MARA, Ministry of Higher Education, Malaysia.

**Institutional Review Board Statement:** Not applicable.

**Informed Consent Statement:** Not applicable.

**Data Availability Statement:** Data can be provided based on the request.

**Acknowledgments:** The authors, i.e., Eni Murdiati, Jawazi Jawazi, Nor Balkish Zakaria, and Kazi Musa would like to thank the support from the Accounting Research Institute HICoE of Universiti Teknologi MARA, Ministry of Higher Education, Malaysia, and Universitas Islam Negeri Raden Fatah, Palembang, Indonesia.

**Conflicts of Interest:** We declare no conflict of interest.

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
