# Peer review of "Does University–Industry Engagement Assist Women in Generating Business Income in Emerging Economies? Evidence from Malaysia"

_economies, doi:10.3390/economies11090239_

Round 1
Reviewer 1 Report (Previous Reviewer 4)
Dear Authors,
I read your paper and your cover letter. I am satisfied with the changes you did to the paper. I also noticed other improvements besides the ones highlighted by me.
Still, I noticed you did not use the template of the journal and did not format the paper accordingly (including references in the text and at the end).
Great success with your research!
Author Response
Reviewer 1:
I read your paper and your cover letter. I am satisfied with the changes you did to the paper. I also noticed other improvements besides the ones highlighted by me.
- a: Still, I noticed you did not use the journal template and did not format the paper accordingly (including references in the text and at the end).
Great success with your research!
Response to Reviewer 1:
Response: 1. a: We follow the journal template, and format of the paper accordingly.
Besides, we have updated the reference according to the journal format (Chicago Style) in both the text and in the end.
Thank you very much for reading our paper. We are happy to hear that you like our study. Once again thank you.

Reviewer 2 Report (New Reviewer)
I very much appreciate the main idea and topic of the research, which is interesting not only for academia but also for business.
The abstract of the article appropriately describes the main results of the paper and entices the reader to read the article.
The keywords are chosen appropriately as well and the title of the paper matches the content.
The only comment here is the failure to mention that the research is in Malaysia.
The literature review is appropriately conceived and has a number of literature sources as well and I appreciate the clear definition of the hypotheses based on the literature. The authors' thoughts are evident in the text.
The methodology section is clearly conceived and contains all the basic information, but I have a few minor comments:
1. It is not clear how large the SMEs in Malaysia are. Here I would recommend to use also the definition of SME from: https://doi.org/10.2478/euco-2019-0033
2. It is not clear from the text why 20km
3. It is not clear how many total SMEs are in the locality and therefore what percentage have been reached.
The whole article is pretty narrowly focused on women's entrepreneurship, but it's not clear from the text why. It would have been useful to address gender more in a literature review, for example.
The results are appropriately handled and are clear and understandable. Just a few comments again:
Tables 1 and 2 are not relevant I recommend to delete.
Variables I recommend to name better, the abbreviations used are not intuitive.
The discussion at the end of the paper is very weak, I recommend including a few more ideas from other authors.
Author Response
Does University-Industry Engagement Assist Women in Generating Business Income in Emerging Economies? Evidence from Malaysia
We are thankful to you for your patience in reading your paper. We try to incorporate your important comments in the manuscript that actually help us to improve our paper.
Thank you
Reviewers 2:
Comment: I very much appreciate the main idea and topic of the research, which is interesting not only for academia but also for business.
Comment: The article's abstract appropriately describes the paper's main results and entices the reader to read the article.
Comment: The keywords are chosen appropriately as well, and the title of the paper matches the content.
Comment: The only comment here is the failure to mention that the research is in Malaysia.
Response: We have mentioned “Malaysia” in the title of the updated manuscript.
Comment: The literature review is appropriately conceived and has a number of literature sources as well and I appreciate the clear definition of the hypotheses based on the literature. The authors' thoughts are evident in the text.
The methodology section is clearly conceived and contains all the basic information, but I have a few minor comments:
Comment 2.1: It is not clear how large the SMEs in Malaysia are. Here I would recommend to use also the definition of SME from: https://doi.org/10.2478/euco-2019-0033
Response 2.1: The SME's definition and size are provided in sub-sections 1.2 para 3, 4 and 5. We also followed the suggested study in the write-up.
Comment 2.2: It is not clear from the text why 20km
Response 2.2: The justifications for the area selection (20km radius) are provided in section 3 (methodology), para 3 (from lines 3 to 7).
Comment 2.3: It is not clear how many total SMEs are in the locality and therefore what percentage have been reached.
Response 2.3: Some information is provided in section 3 (methodology), para 3 (from lines 11 to 13).
Comment 2.4: The whole article is pretty narrowly focused on women's entrepreneurship, but it's not clear from the text why. It would have been useful to address gender more in a literature review, for example.
Response 2.4: The literature section has been updated, and we provide more light on the women and woman-owned SME issues in the whole section.
The results are appropriately handled and are clear and understandable. Just a few comments again:
Comment 2.5: Tables 1 and 2 are not relevant I recommend to delete.
Response 2.5: Since Tables 1 and 2 are irrelevant, we have removed the tables accordingly.
Comment 2.6: Variables I recommend to name better, the abbreviations used are not intuitive.
Response 2.6: The abbreviation of the variables has been updated and we write it well in the manuscript.
Comment 2.7: The discussion at the end of the paper is very weak. I recommend including a few more ideas from other authors.
Response 2.7: The discussion part is updated and provides logical discussion with references.

This manuscript is a resubmission of an earlier submission. The following is a list of the peer review reports and author responses from that submission.
Round 1
Reviewer 1 Report
1. What is the link between women business owners and the survey results? I don't see any women's information in the survey.
2. There are some categories in the survey, e.g., ownership type. It is better to dummy code these categories.
Author Response
Please refer response to the reviewer 1 file attached.

Reviewer 2 Report
The paper has some interest but:
The introduction is not well structured, the ideas are not linked to one another and the objective of the paper is not specified.
The instrument (questionnaire) is not presented.
The variables in eq. (1) are not explained.
Assuming that the instrument had only 8 items (as the variables in Table 2), this is extremely underdeveloped for a scientific paper.
The results in Table 4 are not explained.
The university-industry engagement (which is apparently the topic of the paper) is not actually captured by the instrument.
In the end, while the literature review, discussion etc. can be improved, the instrument appears to be (but it is not presented in the paper) as extremely simplistic.
Author Response
Please refer file attached.

Reviewer 3 Report
Dear Authors,
I read with interest the paper, which focuses on a relevant topic. However, I have several comments.
In the introduction, it is not clear to what contexts data refer to. Furthermore, data are quite outdated and need to be updated.
Furthermore, in the introduction several claims miss references, so I encourage authors to include them (e.g. lines 46-51).
I encourage authors to report in the introduction studies that focus more on the interaction between SMEs and universities. In fact, I found the reference to Nguyen and Nguyen (2020) in lines 125-130 scarcely related to the research topic.
Furthermore, the introduction misses to tie together SMEs, female entrepreneurship, and university role for SMEs.
Last, but not least important, the research question is not explicated in the introduction.
The first three hypotheses are not clear to me: what do the authors mean by “university marketing”? Or “university consultation” This requires further development in the paper as the discussions reported in the literature review do not support fully the comprehension of the hypotheses. I suggest the authors to revise these paragraphs.
The fourth hypotheses requires strong revision. What does “There is a significant association between women's small business revenue across university training” mean? I do not really understand it.
In the methodology, the variables are not explained. Related to my previous comment concerning research questions, it is not clear to me what the variables represents and how they are computed in the analysis.
In the descriptive statistics, it is not clear what do authors mean by “respondents rely on universities to survive their business”. How do SMEs rely on universities? How it is measured? The percentage of revenue generated by the university is a concept that needs to be discussed.
I also do not get the way in which the marketing contribution, along with others, is defined. How can you say that it is minor, significant, etc.? Of course, this limits the comprehension of the results so I encourage authors to revise the paragraph in this sense.
Conclusions do not really discuss the results, therefore more engagement with literature is required. Also, I cannot see a proper discussion centred on the implications on women entrepreneurship, as this is not taken into account at all in the conclusions.
Good luck with your research.
Author Response
Please refer file attached.

Reviewer 4 Report
I have a few recommendations:
1. Check the guidelines for references in the text and for the references at the end of the paper. There is a specific format.
2. Line 336: to survive their business. Rephrase in a clearer way.
3. Check the English of the paper. For example, line 404, The revenue, the university. Also better explain this short statement: Revenue is the SMEs' gross income. The university provides marketing, consulting.... I would add the word services there to be more precise.
4. Before Conclusions, add a Discussion section where you analyse your results in comparison with results of other researchers, no matter if they reached or not the same conclusions.
5. Conclusion section should include: Theoretical and Practical implications of your results (usefulness, for whom, novelty of your research), Limitations of your research and Future research directions.
Author Response
Please refer file attached

Round 2
Reviewer 2 Report
In the previous version of the review, I asked the authors to provide the instrument. The authors have not provided the instrument in the revised version. This is unfortunate, because one very important aspect of science is for the public to have access to the instruments and to be able to replicate the research. Moreover, there is also another dimension to this problem: the fact that the hypotheses (and everything that the authors claim in the results) are not in line with the measurement of the variables. Specifically, the hypotheses refer to: "marketing consultation", "consultancy contribution", "university training". However, the variables do not measure any of these aspects of reality. The variables (as collected by the questionnaire for which do not know the actual items) refer to the "perception" (lines 417-428) of the respondents of any of the aforementioned aspects. There is a conceptual difference between "the effect of university appearance" (whatever that means) and "perception of the effect of university appearance" (measured on a scale of 1 to 4). Therefore, the authors do not know (and do not measure) what is the actual effect of X on Y, because the instrument collects the perception of the respondent on the effect of X on Y. Perceptions are subjective, and the scale is simplistic (as said in the previous version of the review). The authors can claim they found a relationship between revenue and the perception of the effect of training, and not between revenue and university training.